# In Vitro Anti-HSV-1 Activity of Polyphenol-Rich Extracts and Pure Polyphenol Compounds Derived from Pistachios Kernels (*Pistacia vera* L.)

**DOI:** 10.3390/plants9020267

**Published:** 2020-02-18

**Authors:** Maria Musarra-Pizzo, Rosamaria Pennisi, Ichrak Ben-Amor, Antonella Smeriglio, Giuseppina Mandalari, Maria Teresa Sciortino

**Affiliations:** 1Department of Chemical, Biological, Pharmaceutical and Environmental Sciences, University of Messina, Viale SS. Annunziata, 98168 Messina, Italy; mmusarrapizzo@unime.it (M.M.-P.); rpennisi@unime.it (R.P.); ichrak.benamor.etud@fss.usf.tn (I.B.-A.); asmeriglio@unime.it (A.S.); gmandalari@unime.it (G.M.); 2Shenzhen International Institute for Biomedical Research, 140 Jinye Ave. Building A10, Dapeng New District, Shenzhen 518116, China; 3Unit of Biotechnology and Pathologies, Higher Institute of Biotechnology of Sfax, University of Sfax, Sfax 3029, Tunisia

**Keywords:** herpes simplex virus, antiviral, pistachios, polyphenols

## Abstract

Natural compounds are a prominent source of novel antiviral drugs. Several reports have previously shown the antimicrobial activity of pistachio polyphenol extracts. Therefore, the aim of our research was to investigate the activity of polyphenol-rich extracts of natural shelled (NPRE) pistachios kernels (*Pistacia vera* L.) on herpes simplex virus type 1 (HSV-1) replication. The Vero cell line was used to assess the cytotoxicity and antiviral activity. The cell viability was calculated by detection of cellular ATP after treatment with various concentrations of NPRE. For antiviral studies, five nontoxic-concentrations (0.1, 0.2, 0.4, 0.6, 0.8 mg/mL) were tested. Our study demonstrated that treatment with NPRE (0.4, 0.6, 0.8 mg/mL) reduced the expression of the viral proteins ICP8 (infected cell polypeptide 8), UL42 (unique long UL42 DNA polymerase processivity factor), and US11 (unique short US11 protein), and resulted in a decrease of viral DNA synthesis. The 50% cytotoxic concentration (CC_50_), 50% inhibitory concentration (EC_50_), and the selectivity index (SI) values for NPRE were 1.2 mg/mL, 0.4mg/mL, and 3, respectively. Furthermore, we assessed the anti-herpetic effect of a mix of pure polyphenol compounds (NS MIX) present in NPRE. In conclusion, our findings indicate that natural shelled pistachio kernels have remarkable inhibitory activity against HSV-1.

## 1. Introduction

Herpes simplex virus (HSV) infection is quite common in adults and neonates and is associated with oral mucocutaneous lesions and/or genital infections. HSV-1 is the primary cause of neonatal and sporadic encephalitis and, in absence of treatment, the mortality rate associated with central HSV infection is around 70% [1]. Furthermore, the virus persists in the host body and periodically reactivates, especially under certain conditions such as stress and immunosuppression. Although HSV is incurable, acyclovir (ACV) and related nucleoside analogues such as valacyclovir (VCV), famciclovir, and ganciclovir, are widely used for the prophylaxis and treatment of HSV infections [2]. These drugs are considered first-line drug treatments for HSV infections and they have a similar anti-HSV mechanism. However, more than 40% of humans have recurrent infections, and repeated pharmacological treatment often results in the development of drug-resistant HSV strains [3]. Nucleoside analogues are monophosphorylated by the HSV-encoded thymidine kinase (TK) and subsequently converted by the host cell kinases in triphosphate form, which acts as competitive inhibitor of the deoxyguanosine triphosphate for the HSV-encoded DNA polymerase enzyme, thus blocking the viral replication. The mechanism of drug-resistance is due to several mutations occurring in HSV-DNA polymerase or thymidine kinase, and this has become a growing global health problem, especially for immunocompromised patients [4,5,6]. Therefore, the search for new antiviral agents is a main target for many researchers, and natural products have gained much interest because they represent a good source of novel anti-HSV molecules acting with different mechanisms other than nucleoside analogues. A large number of natural products derived from microorganisms, fungi, plants, and animals have already been reported as limiting HSV infections. Among them, crude extracts as well as bioactives isolated from plants, such as plant-derived phenolic compounds, are known for their broad spectrum of antiviral and antioxidant effects [7,8]. To date, several studies reported the anti-HSV activity of phenolic and flavonoid compounds such as caffeic acid [9], curcumin [10,11], epigallocatechin [12], kaempferol [13], emodin [14,15], and glycerine [16]. Recently, the antimicrobial properties of polyphenol-rich fractions derived from pistachios (*Pistacia vera* L.) were reported [17,18], showing that polyphenols from pistachios have bactericidal activity against *Listeria monocytogenes*, *Staphylococcus aureus*, and MRSA (methicillin-resistant *Staphylococcus aureus*) strains. The antiviral activity of lipophilic extracts obtained from different parts of *Pistacia vera* has been screened against HSV and the results indicated that kernel and seed extracts showed significant antiviral activity compared to the rest of the extracts [19]. The goal of the present work was to investigate the effect of polyphenol-rich extracts from natural shelled (NPRE) pistachios kernels (*Pistacia vera* L.) on anti-HSV-1 activity. In addition, we assessed the anti-herpetic effect of a mix of the most abundant pure polyphenol compounds (NP MIX) present in NPRE. Thus, both NPRE and NP MIX were tested against HSV type I by binding assay. The data of the antiviral effect of a polyphenolic mixture obtained from pistachio kernels could be helpful for developing of new formulations for topical use.

## 2. Results

### 2.1. Polyphenolic Composition of the Pistachio Extract

The polyphenols identified in NPRE were previously reported [18]. Briefly, catechin, eryodictiol-7-O-glucoside, gallic acid, and protocatechuic acid were the most abundant compounds, followed by caffeic acid. The total amount of polyphenols was 8.1 mg/100 g fresh weight of NPRE. The identification and quantification of the polyphenols present in NPRE was useful to prepare NP MIX.

### 2.2. Cytotoxicity

To study the cytotoxic effects of pistachio kernels, we performed a first step of experiments by testing polyphenol-rich fractions derived from natural raw shelled (NP). To examine the effect on the growth of cell monolayer, Vero cells were treated with different concentrations of NPRE for 24 h and the cell morphology was then observed. The preliminary screening indicated that treatment with NPRE did not affect the growth of cell culture. The NPRE was then diluted and added to cell culture medium, and Vero cells were incubated with various dilutions (0.4–1.4 mg/mL) for 72 h. DMSO was always maintained below 1% to limit the cytotoxic effect on cells. However, at each dilution, a control sample containing DMSO was included in the assay. The cell proliferation and viability were measured by means of the production of cellular ATP, as described in the Materials and Methods section. The 50% cytotoxic concentration (CC_50_) was calculated from concentration-effect curves by using non-linear regression analysis (Figure 1), and the CC_50_ value was 1.2 mg/mL (Table 1).

### 2.3. Plaque Reduction Assay

The antiviral effect of NPRE was determined against herpes simplex virus type 1 (HSV-1) by using (i) plaque reduction assay, (ii) analysis of viral protein expression, and (iii) quantification of viral DNA. For plaque reduction assay, HSV-1 was serially diluted in medium to reach 50 PFU (plaque forming unit) in 100 µL, and the viral suspension was added on Vero cells and incubated for 1 h at 37 °C. Then, the cells were treated with various non-cytotoxic concentrations (0.1, 0.2, 0.4, 0.6, 0.8 mg/mL) of the NPRE or solvent (DMSO) diluted in medium containing 0.8% methylcellulose. Acyclovir was used as a positive control at the concentration of 20 µM. The plaques were visualized after 3 days using crystal violet staining. The results showed that NPRE exhibited a concentration-dependent antiviral effect, and by using the maximal non-cytotoxic concentration (0.8 mg/mL) plaque formation was totally reduced (100%) as obtained with 20 µM of acyclovir treatment (positive control) (Figure 2). The half-maximal effective concentration (EC_50_) was calculated from concentration-effect curves by using non-linear regression analysis. The EC_50_ value was found at 0.4 mg/mL and the selectivity index (SI) was 3 (Table 1). No antiviral activity was found when the individual polyphenol compounds were used (data not shown).

### 2.4. Analysis of Viral Protein Expression

The results from the plaque reduction assay showed a significant inhibitory activity by adding the NPRE immediately after the attachment step. This suggests that the effect of NPRE might be related to biochemical events that occurred immediately after this time frame, such as gene expression and protein synthesis of the alpha (α), beta (β), and gamma (γ) proteins, which are important for HSV-1 replication. Therefore, in order to investigate the effect of NPRE on the expression of HSV-1 proteins, Western blot analysis was performed. Vero cells were uninfected (mock) or infected with HSV-1 and then treated with non-cytotoxic concentrations of NPRE in the range of the EC_50_ value (0.4, 0.6, and 0.8 mg/mL) for 24 h and DMSO was used in the control samples. The samples were lysed and the total cellular proteins were separated by SDS-PAGE. The optical densitometry analysis of immunoblot band intensity was quantified with TINA imaging analysis software (version 2.10, Raytest, Straubenhardt, Germany) as relative to the band intensity of housekeeping gene glyceraldehyde 3-phosphate dehydrogenase (GAPDH). The analysis revealed that NPRE reduced the accumulation of the tested proteins, particularly the infected cell polypeptide 8 (ICP8) and the unique long UL42 DNA polymerase processivity factor (UL42) β proteins, necessary for viral DNA replication, and the unique short US11 (US11) tegument γ protein. As shown in Figure 3, although all of the three tested proteins were accumulated in HSV-1-infected and untreated Vero cells (lane 2), infected and treated cells accumulated a small amount of viral ICP8, UL42, and US11 proteins (lane 3, lane 4, lane 5, respectively), particularly the samples treated with the maximal non-cytotoxic concentration (0.8 mg/mL; lane 5).

### 2.5. Quantification of Viral DNA

In order to investigate whether the decrease on the aforementioned protein level resulted in a decrease of viral DNA synthesis, we performed a quantitative real-time PCR analysis by using specific TaqMan probe for HSV-1. Vero cells were uninfected (mock) or infected with HSV-1 at multiplicity of infection (MOI) 1 and treated or untreated with 0.6 and 0.8 mg/mL of the NPRE for 24 h. Then, the cellular DNA was extracted and precipitated from the interphase and organic phase and used for real-time PCR analysis. As shown in Figure 4, the accumulation of viral DNA was significantly reduced in the infected and treated samples compared to the untreated infected samples, suggesting that the NPRE treatment might have interfered with the DNA synthesis process, and this resulted in a final decrease of viral DNA synthesis.

### 2.6. Binding Assay Using NPRE and NP MIX

To confirm these data, further investigations were carried out to evaluate the mechanism of action of polyphenols in the process of inhibition of viral replication. The previous data showed that treatment with 0.8 mg/mL of NPRE extract resulted in a 100% reduction in the number of plaques and reduced viral DNA and protein accumulation, compared to the control. Furthermore, treatment with NPRE extracts at the same concentration showed good tolerability as there was no significant cytotoxicity following the treatment. The binding inhibition assay was chosen as it is a useful test to verify whether the binding phase of the infectious process is inhibited by the treatment. The assay was performed using a recombinant virus named as HSV-1-VP26 GFP expressing a GFP (green fluorescent protein)-tagged viral protein 26 (VP26) and by measuring the auto-fluorescence of VP26-tagged protein. The viral suspension was incubated for 1 h at 4 °C with the NPRE (0.8 mg/mL). The infection was conducted at 4 °C, allowing the virus to bind to cellular receptors only, but not to enter the cell. The results from the assay showed that the treatment with NPRE had no effect on the initial attachment step (data not shown).

Consequently, we tried to understand if the antiviral activity was mainly exerted by the more representative polyphenol components present in the NPRE. Indeed recent data obtained in our lab demonstrated that a pure fraction containing the main polyphenol compounds identified in almond skin extracts had both antimicrobial and antiviral activity [20]. On the basis of this, we formulated an NP MIX (0.1 mg/mL) that could have a similar impact on antiviral activity when compared to the polyphenol-rich extracts of natural shelled pistachios. The inhibition of viral replication was evaluated by binding assay using VP26-HSV-1 virus. Figure 5 shows that the treatment with NP MIX resulted in a 50% decrease of the auto-fluorescence of viral protein VP26 (Figure 5a,b) as well as a decreased level of protein expression compared to the control (Figure 5c; lane 3 vs. lane 2). Because NP MIX did not show a good tolerability due to the significant cytotoxicity effect following the treatment, which resulted in 40% cell proliferation index reduction compared to untreated cells (data not shown), further investigations are needed in order to better understand the behaviour and the tolerability of the NP MIX as well as its antiviral activity.

## 3. Discussion

Because no prophylactic vaccine has been found against HSV, the spread of drug-resistant strains has become a major health problem. To note, new evidence indicates an ongoing recombination between HSV-1 and HSV-2 in humans, resulting in increasing genital herpes due to HSV-1 [21,22]. Therefore, herpes viruses have become a potential risk factor for HIV infection in humans, and there is an increased need to find novel drugs to eradicate HSV infections. Over the past decade, crude extracts isolated from plants, as well as pure compounds, have been widely tested for their antiviral activity against HSV infections. Although nucleoside analogues are known to inhibit HSV-DNA replication enzymes, natural compounds can interfere with multiple stages of the virus life cycle, including latency and recurrence of the infections. In addition, besides the study of their bio activity proprieties, the analysis of the structure–activity relationships could provide additional information on their potential antiviral activity and could be useful to develop model of novel anti-herpetic agents. Natural extracts obtained from plants using various extraction approaches have shown a wide spectrum of antiviral activity, including anti-HSV-1 activity. Indeed, natural extract composition that possess phenolic, flavonoid, alkaloid, saponin, esteroide, glycoside, and tannin have been found to interfere with viral replication [8,23,24,25]. In particular, extracts rich in polyphenols have been found to interfere with the viral particles directly through inhibition of virus attachment [26]. In the present study, we found that extracts rich in polyphenols from raw shelled pistachios (NPRE) have a significant antiviral effect against HSV-1. In particular, we demonstrated that the NPRE exhibited a concentration-dependent antiviral effect, and when the maximal non-cytotoxic concentration (0.8 mg/mL) was used, plaque formation was totally reduced compared to 20 µM acyclovir. In addition, these results showed a significant inhibitory activity when NPRE was immediately added after the attachment step, suggesting that NPRE might interfere with the expression of viral proteins by preventing their transcription or translation, and this resulted in a final decrease of viral DNA as well. To note, the pre-treatment of the virus with the NPRE had no effect on the attachment step (data not shown), confirming a potential antiviral mechanism related to events that occurred immediately after the binding of the virus, such as gene expression and DNA synthesis. Because several unauthenticated crude extracts of plants were observed as exhibiting anti-HSV-1 activity, we tried to identify specific compounds that were responsible for this activity. Indeed, pre-treatment of the virus with a standard NP MIX (0.1 mg/mL), containing the most representative polyphenols in NPRE, led to a significant reduction of the expression of VP26 GFP viral protein, resulting in a decreased infectivity of the virus. These findings revealed a novel potential therapeutic effect of a polyphenols mix from NPRE. Given their lower cellular tolerability (data not shown), further investigation should be carried out to study the antiviral and cytotoxic effect and to define the mechanism of action of the polyphenol mixture as well as to establish their CC_50_, EC_50_, and SI. However, the antiviral activity of NPRE is likely due to the higher content of polyphenols and their possible synergistic interactions, especially hydroxybenzoic acids, catechin, and isoquercetin, which were the main abundant polyphenols in the pistachio extracts. Indeed, the bactericidal activity of NPRE was also previously assessed against *Listeriamonocytogenes*, *Staphylococcus aureus*, and MRSA strains [17], and recently, simulated human digestion analyses showed that polyphenols from pistachios are bioaccessible in the upper gastrointestinal tract [27]. However, it seems that compounds not considered in the preparation of NP MIX play a different yet specific role in the inhibition of HSV replication.

## 4. Materials and Methods

### 4.1. Plant Material, Extraction, and NP MIX Preparation

American natural shelled (NP) pistachio kernels were kindly provided by Di Bartolo S.r.l., Calatabiano (Italy). NPs were ground to fine powder as previously reported [18] and polyphenol-rich extracts (NPRE) were obtained following the method reported by Mandalari et al. 2013 [27].

The NP MIX was prepared in DMSO (stock solution 100 mg/mL) according to the natural shelled (NP) polyphenol extract composition, and properly diluted [17]. The most abundant compounds (HPLC-grade, purity ≥ 99%) identified and quantified in the previous work (catechin, eriodictyol-7-*O*-glucoside, gallic acid, protocatechuic acid, caffeic acid, rutin, and isoquercetin) were acquired from Extrasynthese (Genay, France) and mixed in the same proportion in which they were presented into the NP extract (30:20:15:15:11:5:4, w/w).

### 4.2. Cell Culture and Virus

African green monkey kidney cell line (Vero) was cultured in Eagle’s minimum essential medium (MEM; Corning, Corning, NY, USA) supplemented with 6% foetal bovine serum (FBS; Euroclone), 100 U/mL penicillin, and 100 μg/mL streptomycin. The cells were maintained at 37 °C under 5% CO_2_ atmosphere. The HSV-1 laboratory strain (F strain), used for the in vitro experiments was kindly provided by Professor Bernard Roizman (University of Chicago, Chicago, IL, USA). For binding assay, a recombinant virus (HSV-1-VP26GFP) expressing a GFP-tagged VP26 protein was used as described previously [26]. HSV-1 stocks were prepared from Vero cells infected with HSV-1 at low multiplicity of infection (MOI) and harvested at 24 h post infection (p.i.). The cell-free supernatant containing the viral suspension was stored at −80 °C and used for the experiments. Vero cell lines were originally obtained from the American Type Culture Collection (ATCC).

### 4.3. Cell Viability to Test the Cytotoxicity of NPRE and NP MIX

Approximately 2*10^4^ cells/well of Vero cells were cultured in a 96-well plate and incubated with NPRE (0.4, 0.5, 0.6, 0.8, 1, 1.2, 1.4 mg/mL) and NP MIX (0.1 mg/mL) separately for 72 h. A control of the solvent DMSO was included for each dilution. The cell viability was evaluated as production of cellular ATP by using the ViaLight plus kit (Lonza, Basel, Switzerland) according to the manufacturer’s instructions. The cells were lysed with the cell lysis reagent for at least 10 min and then the ATP monitoring reagent plus (AMR plus) was added to each well for 2 min at room temperature (RT). The emitted light intensity related to ATP concentration was measured using a GloMax Multi Microplate Luminometer (Promega Corporation, Madison, WI, USA) and the luminescence value was converted into the cell proliferation index (%) as described previously [26].

### 4.4. Plaque Reduction Assay

Vero cells (4*10^5^ cells/well) were cultured in 12-well plates and incubated with 50 PFU (plaque forming unit) of HSV-1. The viral suspension was adsorbed on cells for 1 h at 37 °C. Next, the inoculum was removed and the NPRE or solvent (DMSO), diluted in medium containing 0.8% methylcellulose, was added at various concentrations (0.1, 0.2, 0.4, 0.6, 0.8 mg/mL). Acyclovir was used as positive control at the concentration of 20 µM. After 3 days, the virus plaques formed on Vero cell monolayers were visualized and counted by crystal violet staining at 10× magnification with an inverted microscope (Leica DMIL).

### 4.5. Binding Assay to Test the Antiviral Activity of NPRE and NP MIX

The binding assay was performed at 4 °C in order to allow the binding of the virus to the cell receptors, but not the entry as previously reported [26]. Briefly, the viral suspension was incubated with the NPRE (0.8 mg/mL) and the NP MIX (0.1 mg/mL) for 1 h at 4 °C. Vero cells were then infected with the virus inoculum at multiplicity of infection (MOI) of 1, and the infection was performed at 4 °C with gentle shaking for 1 h. The plates were then washed 3 times with cold phosphate-buffered saline (PBS) solution to remove the unbound virus particles, and fresh media was then added. After 24 h post infection (p.i.), the expression of the VP26 GFP protein was detected by (i) detection of the auto-fluorescence of VP26-tagged protein and (ii) Western blot analysis. For fluorescent microscopy analysis, the samples were layered on polylysinated slides, fixed with 4% paraformaldehyde (PFA 4%), washed three times with PBS 1X, and stained with Hoechst 33342. Samples were analyzed on a fluorescence microscope (Leitz, Wetzlar, Germany). For Western blot analysis, cellular proteins were extracted using SDS sample buffer 1X and analysed as reported below.

### 4.6. Western Blot Analysis

Vero cells (2*10^5^ cells/well) were cultured in 24-well plates and uninfected (mock) or infected with HSV-1 (F) at MOI 1. The virus was adsorbed on cells for 1 h at 37 °C and then removed. The cells were treated for 24 h with different concentrations of NPRE (0.4, 0.6, 0.8 mg/mL). A control of the solvent (DMSO) was also included. Cellular proteins were extracted using SDS sample buffer 1X (62.5 mM Tris-HCl pH 6.8; Dithiothreitol (DTT) 1 M; 10% glycerol; 2% SDS; 0.01% Bromophenol Blue). Equal quantities of proteins were resolved by SDS-10% polyacrylamide gel electrophoresis (PAGE) and transferred to nitrocellulose membranes (BioRad Life Science Research, Hercules, CA, USA). After blocking, the membranes were probed overnight at 4 °C with specific antibodies for HSV-1 β proteins ICP8 and UL42, and γ protein US11. Specific proteins were detected using secondary anti-mouse antibody linked to horseradish peroxidase (HRP). The chemiluminescence was detected by Western HRP substrate (Merk, Millipore, Burlington, MA, USA). The housekeeping GAPDH and β-actin proteins were used as loading control. In the experiment using NP MIX, the Vero cells were treated for 24 h with 0.1 mg/mL and processed as described above. The membrane was probed with a specific antibodies for GFP proteins, and chemiluminescence was detected by Western HRP substrate (Merk, Millipore).

### 4.7. Antibodies

Anti-GAPDH (sc-32233), anti-GFP (sc-9996), and anti-UL42 (sc-53333) antibodies were purchased from Santa Cruz Biotechnology (Santa Cruz, CA, United States). Anti-β-actin (ab8226) antibody was from Abcam.Monoclonal antibodies against US11 and ICP8 were provided by Professor Bernard Roizman. Goat anti-mouse immunoglobulin G (IgG) antibody, HRP conjugate was purchased from Merk, Millipore.

### 4.8. DNA Extraction and Real-Time PCR

Cellular DNA was extracted from Vero cells using TRIzol (Life Technologies, Carlsbad, CA, United States), according to the manufacturer’s instructions. Vero cells (2*10^5^ cells/well) were cultured in 24-well plates and uninfected (mock) or infected with HSV-1 (F) at MOI 1 for 1 h at 37 °C. The cells were then treated with NPRE (0.6 and 0.8 mg/mL) for 24 h. The solvent (DMSO) was used in the control samples. The DNA solutions were extracted with phenol-chloroform and precipitated from the interphase and organic phase with 100% ethanol. The DNA pellet was washed twice in a solution containing 0.1 M sodium citrate in 10% ethanol and then dissolved in 8 mM NaOH. The concentration of DNA was determined by fluorometer analysis with the Qubit double stranded DNA (dsDNA) HS (High Sensitivity) Assay Kit according to the manufacturer’s instructions. Quantitative real-time PCR was performed in a Cepheid SmartCycler II System (Cepheid Europe, Maurens-Scopont, France), using specific TaqMan probe. Total cellular DNA (1 µg) was mixed with 0.5 μM of each forward and reverse primers, 1 µM of TaqMan probe, 1 µM of deoxyribonucleotide triphosphate (dNTP) mix, NH_4_ reaction buffer 1X, 2 mM of MgCl_2_, and 5 U/µL of thermostable DNA polymerase BIOTAQ (BIO-21040 Bioline) in a total volume of 25 µL. The oligonucleotide primer pairs were as follows: HSV-1 Fw 5’-catcaccgacccggagagggac; HSV-1 Rev 5’gggccaggcgcttgttggtgta, HSV-1 TaqMan probe 5’-6FAM-ccgccgaactgagcagacacccgcgc-TAMRA, (6FAM is 6-carboxyfluorescein and TAMRA is 6-carboxytetramethylrhodamine). The amplification was performed following specific steps (10 min at 95 °C, 30 s at 95 °C for 40 cycles, 30 s at 55 °C, 30 s at 72 °C, 5 min at 72 °C) and a negative sample was used as amplification control for each run. The relative quantitation of HSV-1 DNA was generated by comparative C_t_ method using GAPDH as housekeeping gene.

### 4.9. Statistical Analysis

The data analysis and the graphical representations were performed using the GraphPad Prism 6 software (GraphPad Software, San Diego, CA, USA). Student’s *t*-test was used for the statistical analysis of the data presented as means of three independent experiments ± standard deviations (SD). The asterisks (*, **, and ***) indicate the significance of *p*-values less than 0.05, 0.01, and 0.001, respectively. The 50% cytotoxic concentration (CC_50_) and half maximal effective concentration (EC_50_) were calculated from concentration-effect curves by using non-linear regression analysis. Quantitative densitometry analysis of immunoblot band intensity was performed with TINA software (version 2.10, Raytest, Straubenhardt, Germany).

## 5. Conclusions

Our current finding indicates that polyphenols from pistachios are effective against HSV-1. In conclusion, we can assume that the antiviral effects of NPRE are the result of a balance of the individual polyphenolic components that in combination exert the anti-viral activity. Therefore, pistachio polyphenols could become great candidates for the development of novel topical or oral drug formulations for the treatment of HSV-1 infections, either alone or in combination with standard antiviral therapies. However, further studies will be helpful in studying the mechanisms of antiviral activity as well as to identify bioactive compounds responsible for the anti-HSV activity. Thus, amongst other natural products, pistachio extracts could provide a novel treatment against HSV-1 infections, as well as a novel strategy to overcome problems related to drug-resistant strains. The results present here agree with a previous study, in which polyphenols were found to possess, beyond anti-inflammatory properties, antiviral activity also, thus suggesting possible further interest for the pistachio product waste as a source of a new pharmaceutical agent.

## Figures and Tables

**Figure 1 plants-09-00267-f001:**
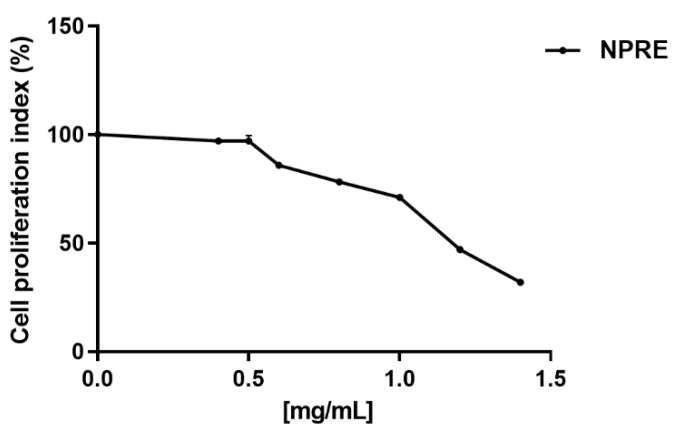
Cell viability of Vero cells treated with increasing concentration of polyphenol-rich extracts from natural shelled pistachios kernels (NPRE). The cell proliferation index (%) was calculated by means of cellular ATP level measured after 72 h treatment. The values were expressed as percentages of treated vs. control cells (DMSO). Each value is the mean ± standard deviation (SD) of three experiments.

**Figure 2 plants-09-00267-f002:**
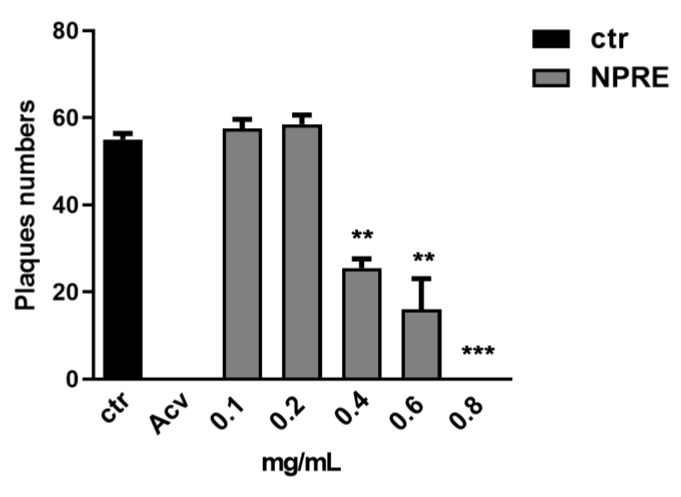
Effects of NPRE on herpes simplex 1 (HSV-1) replication by plaque reduction assay. Vero cells were infected with HSV-1 (50 plaque forming units (PFU)/100 µL) for 1 h and then treated with NPRE (0.1, 0.2, 0.4, 0.6, 0.8 mg/mL). The DMSO was used in the control samples and acyclovir (20 µM) was used as positive control. Data are expressed as a mean (± SD) of at least three experiments, and asterisks (**, and ***) indicate the significance of *p*-values less than 0.01 and 0.001, respectively.

**Figure 3 plants-09-00267-f003:**
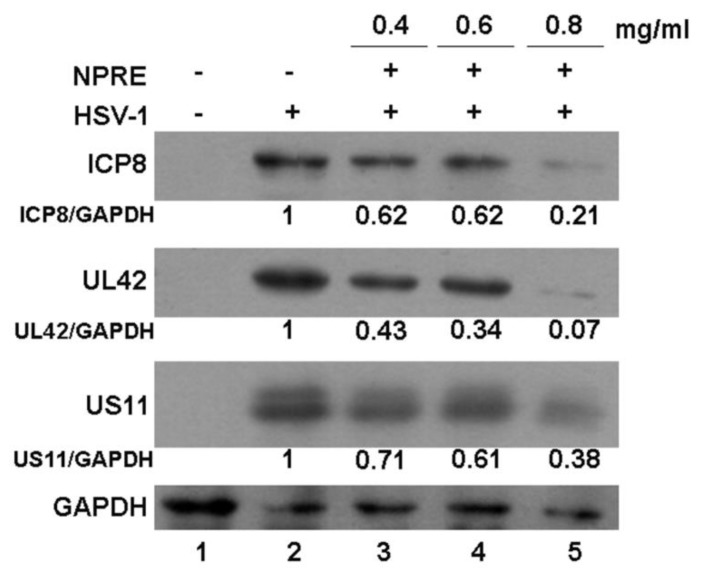
Analysis of viral protein expression by Western blot analysis.

**Figure 4 plants-09-00267-f004:**
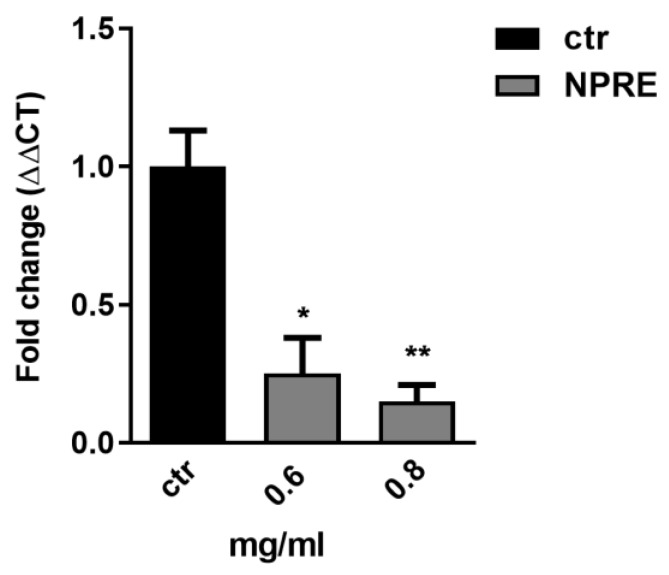
Relative quantization of viral DNA with specific HSV-1 TaqMan probe in real-time PCR. DMSO was used in the infected control samples. Data are expressed as a mean (± SD) of at least three experiments and asterisks (* and **) indicate the significance of *p*-values less than 0.05 and 0.01, respectively.

**Figure 5 plants-09-00267-f005:**
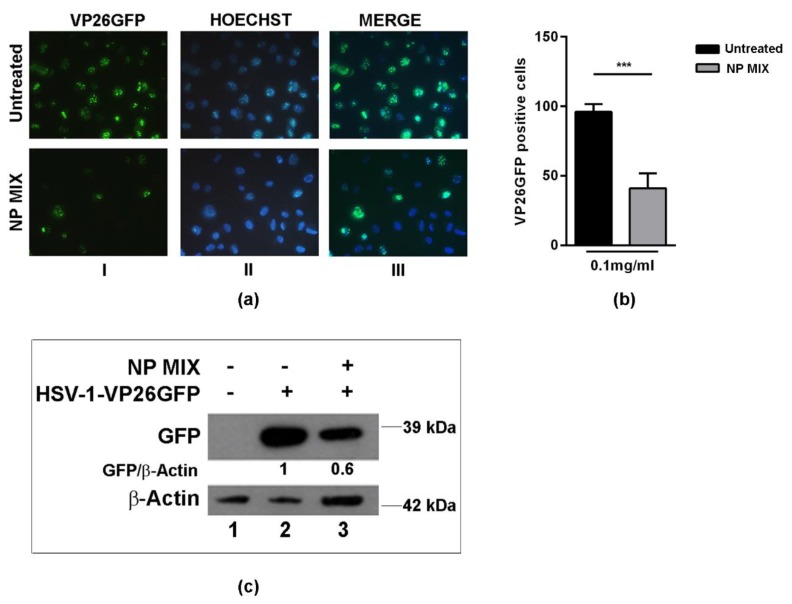
Effect of the NP MIX treatment on HSV-1 replication. Vero cells were either infected or mock-infected with HSV-1-VP26 GFP (green fluorescent protein-tagged capsid protein VP26) at multiplicity of infection (MOI) 1, as described in the Materials and Methods section. Then, the cells were analysed at 24 h post infection (p.i.): (**a**) fluorescent images showed the green dots representing the VP26 GFP viral antigen localization (I)—Hoechst (blue) was used to stain the nuclei (II) and the merged images are shown in column III; (**b**) the graph is indicative of the percentage of VP26 GFP-positive cells; (**c**) Western blot analysis of VP26 GFP-tagged protein. Data are expressed as a mean (± SD) of at least three experiments, and asterisks (***) indicate the significance of *p*-values less than 0.001.

**Table 1 plants-09-00267-t001:** Selectivity index (SI), cytotoxicity (CC_50_), and antiviral activity (EC_50_) of polyphenol-rich extracts from natural shelled pistachios kernels (NPRE).

Extracts	CC_50 (mg/mL)_ ^a^	EC_50 (mg/mL)_ ^b^	SI ^c^
NPRE	1.2	0.4	3

^a^ 50% cytotoxic concentration; ^b^ 50% inhibitory concentration; ^c^ ratio of CC_50_ to EC_50_.

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
