# Peer review of "In Vitro Anti-HSV-1 Activity of Polyphenol-Rich Extracts and Pure Polyphenol Compounds Derived from Pistachios Kernels (*Pistacia vera* L.)"

_plants, 2020, doi:10.3390/plants9020267_

Round 1
Reviewer 1 Report
The manuscript entitled “In vitro anti-HSV-1 activity of polyphenol-rich extracts derived from pistachios kernels (Pistacia vera L.)” deals with the study of an alternative therapeutic treatment (to the standard Acyclovir and related nucleoside analogues) against Herpes Simplex Virus infections in order to reduce the risk of drug-resistance.
Overall, it is well written and organized. The experiments are correctly set and R&D appeared coherent. Anyway, my main concern regards the “study on the structure-activity relationships” that the authors invoke in the text (Line 211). I mean that it is not clear if and how the authors attribute the antiviral activity of polyphenols-rich extract from Pistachio to its particular composition. Reading the text, the effect seems to be due to the total amount of polyphenols in the extract; it would mean that starting from Pistachio or other source of PP should be the same, is it? Moreover, comparing the real extract to standard mix (another point a bit unclear!), did the authors determine whether the activity is from an additive or synergic effect of polyphenols compounds?
Other specific comments are listed in the attached file plants-703902_R1.

Author Response
Point-by-point response is provided in the attacched file

Reviewer 2 Report
The manuscript entitled “In vitro anti-HSV-1 activity of polyphenol-rich extracts derived from pistachios kernels (Pistacia vera L.).” by Maria et al., describes here the antiviral activity of natural shelled pistachios kernels with remarkable inhibitory activity against HSV-1.
Although authors state here an effective approach, there are few points’ needs to be addressed or included in the study before accepting this manuscript for publication.
General points:
More citations should be provided by authors.
Manuscript is very confusing and not easy to follow, so authors are requested to present it clearly while resubmitting the MS.
Specific points:
Abstract: line 24: “For antiviral studies 5 nontoxic-concentrations” change to “For antiviral studies, five nontoxic-concentrations”
Introduction: line 37: “treatment, the mortality rate is around 70%.- Provide citation
Line 41: Provide citation.
Line 43-44: “treatment often results in the spread of drug-resistant HSV strains” change to “treatment often results in the development of drug-resistant HSV strains”
Line 69: NPRE- write it in full and abbreviate later. It was abbreviated in abstract but should be introduced once in the main text too.
The total amount of polyphenols was 8.1 mg/100 g fresh weight of NPRE- Was it done in current work? If so, its lacking in Methodology section and if not section 2.1 should be removed completely as it is reluctant and don’t provide any data from current work.
Authors are mentioning always “To study the cytotoxic effects of pistachios kernels we performed a first step of experiments by testing polyphenol-rich fractions derived from natural raw shelled (NP).” Which is not true as methodology states that the test was done by cocktail prepared using synthetic standards. Eventhough, authors maintained the ratio of the polyphenols depending on previous report, you cannot present as fraction derived from plant.
Authors are requested to make it very clear from starting that the activity was done using chemical standard not using the natural source.
Even the title is missleading in that case and hence should be changed.
It is also advised to compare the results with natural fraction rice with these polyphenols and then also compare it with individual polyphenols to know the synergistic effect of these polyphenols.
Author Response

(The authors gave the same response as above.)

Reviewer 3 Report
In this study, the authors evaluated a crude pistachio kernel extract on HSV in vitro. Overall an interesting study and the following suggestions may further improve the manuscript.
Abstract: No need to add long background. Abstract should be concise and showcase major findings with values and statistics and end with a practical implication of the study.
Line 24: how these 5 “nontoxic” concentrations were determined.
Line 27: CC50, EC50, SI must be elaborated when used for the first time.
Lines 63-65: Authors must discuss whether there is other literature available on the subject matter and if so, how their study may be novel.
Figure1: Authors calculated cell viability in terms of cell proliferation index, how this unit translates into viable cell number (measured with trypan blue or other dye-based counting methods).
Is there any information on in vivo cytotoxicity caused by NPRE?
In addition, it would be interesting to know whether the plant extract show comparable efficacy against multiple strains of HSV including acyclovir resistant HSV strains.
Author Response
point-by-point response is provided in the attacched file

Reviewer 4 Report
The authors examined the effects of pistachios kernels (Pistacia vera L.) extracts on anti-HSV-1 activity. Please avoid materials and methods in the results. The discussion part is also very preliminary, with the results obtained not being discussed satisfactorily. Please improve the quality of Figure 5a.
Author Response

(The authors gave the same response as above.)

Round 2
Reviewer 1 Report
I am enough satisfied by answers and comments from the authors to my concerns. Therefore, now I am considering the article suitable for the publication in the journal.
Just few minor points are listed in the attached file plants-703902_R2.

Author Response
According with the reviewer request we modified the text

Reviewer 2 Report
The manuscript has been improved a lot after taking in consideration of reviewers suggestion and hence, I recommend the manuscript for acceptance.
Author Response

(The authors gave the same response as above.)

Reviewer 4 Report
The authors improved the manuscript according to the reviewer's comments and thus I recommend this article for publication in Plants.
Author Response

(The authors gave the same response as above.)
